# Gabapentin Disrupts Binding of Perlecan to the α_2_δ_1_ Voltage Sensitive Calcium Channel Subunit and Impairs Skeletal Mechanosensation

**DOI:** 10.3390/biom12121857

**Published:** 2022-12-12

**Authors:** Perla C. Reyes Fernandez, Christian S. Wright, Adrianna N. Masterson, Xin Yi, Tristen V. Tellman, Andrei Bonteanu, Katie Rust, Megan L. Noonan, Kenneth E. White, Karl J. Lewis, Uma Sankar, Julia M. Hum, Gregory Bix, Danielle Wu, Alexander G. Robling, Rajesh Sardar, Mary C. Farach-Carson, William R. Thompson

**Affiliations:** 1Department of Physical Therapy, School of Health and Human Sciences, Indiana University, Indianapolis, IN 46202, USA; 2Department of Chemistry and Chemical Biology, School of Science, Indiana University, Indianapolis, IN 46202, USA; 3Department of Diagnostic and Biomedical Sciences, School of Dentistry, The University of Texas Health Science Center at Houston, Houston, TX 77054, USA; 4Department of Bioengineering, George R. Brown School of Engineering, Rice University, Houston, TX 77005, USA; 5Department of Medical and Molecular Genetics, School of Medicine, Indiana University, Indianapolis, IN 46202, USA; 6Meinig School of Biomedical Engineering, Cornell University, Ithaca, NY 14853, USA; 7Department of Anatomy, Cell Biology and Physiology, School of Medicine, Indiana University, Indianapolis, IN 46202, USA; 8Division of Biomedical Science, College of Osteopathic Medicine, Marian University, Indianapolis, IN 46222, USA; 9Departments of Neurosurgery and Neurology, School of Medicine, Tulane University, New Orleans, LA 70112, USA

**Keywords:** perlecan, voltage-sensitive calcium channels, α_2_δ_1_, gabapentin, mechanosensation, bone, osteocytes

## Abstract

Our understanding of how osteocytes, the principal mechanosensors within bone, sense and perceive force remains unclear. Previous work identified “tethering elements” (TEs) spanning the pericellular space of osteocytes and transmitting mechanical information into biochemical signals. While we identified the heparan sulfate proteoglycan perlecan (PLN) as a component of these TEs, PLN must attach to the cell surface to induce biochemical responses. As voltage-sensitive calcium channels (VSCCs) are critical for bone mechanotransduction, we hypothesized that PLN binds the extracellular α_2_δ_1_ subunit of VSCCs to couple the bone matrix to the osteocyte membrane. Here, we showed co-localization of PLN and α_2_δ_1_ along osteocyte dendritic processes. Additionally, we quantified the molecular interactions between α_2_δ_1_ and PLN domains and demonstrated for the first time that α_2_δ_1_ strongly associates with PLN via its domain III. Furthermore, α_2_δ_1_ is the binding site for the commonly used pain drug, gabapentin (GBP), which is associated with adverse skeletal effects when used chronically. We found that GBP disrupts PLN::α_2_δ_1_ binding in vitro, and GBP treatment in vivo results in impaired bone mechanosensation. Our work identified a novel mechanosensory complex within osteocytes composed of PLN and α_2_δ_1_, necessary for bone force transmission and sensitive to the drug GBP.

## 1. Introduction

Osteocytes reside deep within the mineralized matrix of bone and have long dendrite-like processes that run through microscopic channels called canaliculi [1]. As osteocytes are uniquely positioned in the bone matrix to communicate with other bone cell types via paracrine signaling and through direct contact with the cellular processes, they are considered the primary mechanosensory skeletal cells [1]. Transmission of mechanical force from the bone matrix to the osteocyte cell membrane was initially thought to occur via direct sensing of whole-tissue strains on the osteocyte surface. However, strains applied to whole bone in vivo during normal locomotion are typically between 0.04–0.3% [2,3], an order of magnitude smaller than the strain necessary to elicit a biochemical response at the osteocyte plasma membrane (1–10%) [4,5,6]. Thus, a mechanism other than direct force transmission from the bone matrix must account for the ability of osteocytes to perceive mechanical input. 

The pericellular space between (PCS) the bone matrix and the osteocyte plasma membrane contains non-mineralized extracellular matrix molecules, including proteoglycans, which are collectively termed the pericellular matrix (PCM) [7]. To explain the mechanism by which tissue-level mechanical strains are transmitted into biochemical responses in osteocytes, the presence of matrix-based “tethering elements” (TEs) able to span the PCS and anchor the osteocyte processes to the mineralized matrix was proposed [8]. This theoretical model was followed by ultrastructural studies using electron microscopy that visually revealed the tethering elements within the PCS [7]. However, the molecular identity of these TEs remained unknown.

Using immunostaining and immunogold assays, we showed that the large heparan sulfate proteoglycan perlecan (HSPG2, PLN) is expressed along osteocyte cell bodies and dendritic processes in cortical bone but not within the mineralized matrix [9]. Furthermore, PLN-deficient mice had fewer TEs within osteocyte canaliculi [9], lower canalicular drag forces, and decreased responses to exogenous loading [10]. Together, these studies identified PLN as a component of the tethering complex in osteocytes, necessary for anabolic responses to mechanical loading. While this finding helped explain how force is transmitted to the osteocyte cell membrane, the PLN-containing tethers must attach to the cell surface to induce biochemical responses.

Intracellular calcium (Ca^2+^) influx is a potent signal in response to force [11]. Ca^2+^ influx is regulated by voltage sensitive Ca^2+^ channels (VSCCs), and in vitro and in vivo studies have shown that VSCCs are necessary for anabolic responses to skeletal loading [12,13]. As PLN deficiency impairs mechanically induced Ca^2+^ signaling in bone [14], we hypothesized that PLN tethers bind VSCC ectodomains, forming what we call a matrix-channel tethering complex (M-CTC), and that this interaction facilitates intracellular Ca^2+^ influx in response to mechanical force. VSCCs are integral membrane proteins composed of the pore-forming α_1_ subunit, which enables calcium (Ca^2+^) entry, and auxiliary subunits including α_2_δ, β, and γ [15] (Figure 1). 

While the pore-forming (α_1_) subunit enables Ca^2+^ entry across the membrane, auxiliary subunits influence gating kinetics of the channel pore. In particular, the α_2_δ_1_ subunit is anchored in the plasma membrane, with the majority of the protein positioned extracellularly, an optimal location to interact with extracellular molecules, such as PLN-containing tethering elements. Interestingly, the α_2_δ_1_ subunit is the binding site of the antiepileptic and neuropathic pain drug gabapentin (GBP) [20,21] (Figure 1). Chronic GBP use is associated with increased fracture risk in humans [22] and impaired bone formation in both human and animal studies [23,24]. However, the mechanism(s) mediating GBP-associated adverse skeletal effects are unclear. Thus, in addition to establishing if PLN directly binds the α_2_δ_1_ subunit of VSCCs, we sought to determine if GBP interferes with binding of the PLN/α_2_δ_1_ complex.

## 2. Materials and Methods

### 2.1. Cell Culture and Immunofluorescence

Immunofluorescence experiments were performed using the murine osteocyte-like cell line MLO-Y4 [25], a generous gift from Dr. Lynda Bonewald (Indiana University School of Medicine, Indianapolis, IN, USA). Approximately 1000 MLO-Y4 cells were seeded onto collagen-I coated 8-well chambers (NUNC™, Rochester, NY, USA) and cultured as described previously [26]. When cells were 80–90% confluent, media was removed, cells were washed with Tris-buffered saline (TBS) and fixed with paraformaldehyde (4%, *v/v*) diluted in TBS for 45 min at room temperature (RT). Cells were washed with TBS to remove residual fixative. To block non-specific binding sites, cells were incubated for 1h at RT in blocking buffer containing BSA (3%, *v/v*) and normal goat serum (10 %, *v/v*) (the source species for the secondary antibodies) diluted in TBS. Cells were incubated with the appropriate primary antibodies (Abs) diluted in blocking buffer for 1h at RT. For co-localization experiments, where association between Ca_v_3.2 (α_1H_), α_2_δ_1,_ and Perlecan (PLN) were performed in osteocytic cells, the following primary Abs were used: affinity-purified rabbit anti-Ca_v_3.2 (α_1_H) polyclonal antibody (1:100) was raised against a synthetic peptide sequence and prepared for our laboratory commercially by ResGen (Invitrogen, Carlsbad, CA, USA), as described [27]. Staining for α_2_δ_1_ was performed as previously reported [28], affinity purified rabbit anti-α_2_δ_1A_ isoform polyclonal antibody (1:500) produced by Bethyl Laboratories (Montgomery, TX, USA) [29] was used. For PLN staining, cells were incubated with rat monoclonal anti-PLN domain-IV (A7L6) primary antibody (1:40) (Abcam, Boston, MA, USA). Following incubation with the primary Abs, cells were washed with blocking solution and incubated with species-specific Alexa Fluor 488 and 555 conjugated secondary Abs (1:200) (Invitrogen, Carlsbad, CA, USA) and DRAQ5™ nuclear stain (1:1000) (Biostatus, Ltd., Shepshed Leicestershire, UK) diluted in blocking solution. To visualize cell membrane glycoproteins, cells were stained with fluorescein conjugated wheat germ agglutinin (WGA) (Invitrogen, Carlsbad, CA, USA). Samples were washed with TBS, mounted, and stored at 4 °C until imaged. Negative controls for cultured cells were performed using non-immune IgGs diluted at concentrations equivalent to primary Abs or without primary Abs. For cells with WGA-FITC staining, N, N′, N″- triacetylchitotriose, the sugar to which WGA binds, was used as a negative control to preabsorb prior immunofluorescence staining. Samples were imaged with a laser scanning microscope (LSM) 510 visible spectrum (VIS) confocal microscope using a 40X C-apochromat water immersion objective (NA 1.2) (Carl Zeiss, AG, Jena, Germany).

### 2.2. Co-Immunoprecipitation and Western Blotting

To determine if α_2_δ_1_ associates with PLN, co-immunoprecipitation assays were performed. MLO-Y4 cells (~90% confluent) cultured on 100 mm dishes were exposed to 500 μL of radio-immunoprecipitation assay (RIPA) lysis buffer containing a protease inhibitor cocktail added just prior to cell lysis (1:100) (Sigma-Aldrich, St. Louis, MO, USA). Plates were incubated with lysis buffer at 4 °C for 1 min. Lysates were scraped from each plate and placed in 1.5 mL tubes. Samples were sonicated and centrifuged (14,000× *g*) for 10 min at 4 °C. Protein concentration was determined using the Pierce BCA protein assay kit (ThermoFisher Scientific, Waltham, MA, USA). Samples were diluted in RIPA buffer to achieve equal protein concentrations. Pre-cleared lysates were added to 100 μL of magnetic Dynabeads (Invitrogen, Carlsbad, CA, USA) complexed to 5 µg of monoclonal anti-PLN A7L6 antibody (Abcam, Boston, MA, USA) or Rat IgG. Lysates and beads were incubated on a rotator at RT for 45 min. The Dynabead-Ab-Ag complex was washed three times with 1× phosphate-buffered saline (PBS). Beads then were resuspended in Laemmli buffer containing β-mercaptoethanol (2%, *v/v*) and boiled for 10 min to release proteins from beads. Lysates were removed from the dynabeads and Western blotting was performed as described [30]. Equal volumes of each sample (20 µL) were electrophoresed in 8–12% Tris-Acetate gels and probed with the anti-α_2_δ_1A_ (Bethyl Laboratories, Montgomery, TX, USA) and anti-PLN A7L6 (Abcam, Boston, MA, USA) primary antibodies (1:500). Blots were probed for β-actin Ab (Cell signaling, Danvers, MA, USA) (1:500) as a loading control. 

### 2.3. Recombinant α_2_δ_1_ Polypeptides

The α_2_ portion of the human α_2_δ_1_ protein (National Center for Biotechnology Information, NCBI, reference sequence NP_00713.2) was produced by GenScript Protein Expression and Purification Services (GenScript Corp, Piscataway, NJ, USA). Briefly, the α_2_ target DNA sequence was designed, optimized, and synthesized by sub-cloning into a plasmid complementary DNA (pcDNA) 3.4 vector and transfection-grade plasmid was maxi-prepared for cell expression. Expi293F^TM^ cells (derived from the human embryonal kidney 293 [HEK 293] cell line), were grown in serum-free Expi293F Expression Medium (ThermoFisher Scientific, MA, USA). Cells were maintained in Erlenmeyer flasks (Corning, NY, USA) at 37 °C with CO_2_ (8% *v/v*) on an orbital shaker (VWR Scientific, Radnor, PA, USA). One day before transfection, cells were seeded at an appropriate density in flasks. On the day of transfection, DNA and transfection reagent were mixed at an optimal ratio and added to the cells. The recombinant plasmid encoding the target protein was transiently transfected into Expi293F cells. Culture supernatants, collected on day 6, were used for protein purification. Conditioned media was centrifuged, filtered, then passed through a HisTrapTM FF crude affinity purification column at an appropriate flowrate. After washing and elution with appropriate buffers, the eluted fractions were pooled, and buffer exchanged to the final formulation buffer. Purified protein was analyzed by Western blot to confirm the molecular weight and purity. The concentration was determined by Micro-Bradford assay with bovine serum albumin (BSA) as a standard (ThermoFisher Scientific, MA, USA). Purified protein was stored in 1× PBS (pH 7.2), filter sterilized (0.22 μm), and packaged aseptically at a concentration of 37 μg/mL.

### 2.4. Recombinant Perlecan Proteins

#### 2.4.1. Full-Length Perlecan

PLN is a large proteoglycan composed of five domains [31]. Full-length PLN was isolated and purified from HT-29 human colorectal cancer cells (formerly called WiDr) (American Type Culture Collection, ATCC, Manassas, VA, USA) as reported [32,33]. These cells were selected because they are able to synthetize large amounts of PLN [34]. Briefly, cells were cultured and maintained in Eagle’s minimal essential medium (EMEM) (ATCC^®^ 30-2003), fetal bovine serum (FBS) (10%, *v/v*) and penicillin/streptomycin (1% *v/v*). When cells were 100% confluent, fresh media with lower FBS levels (2%, *v/v*) was added and conditioned medium was collected. The conditioned medium was filtered and concentrated using the Amicon stirred cell pressure-driven filtration device (400 mL, 46 mm membrane filter) (Millipore, Sigma, Burlington, MA, USA). The resulting high molecular weight concentrated solution was subjected to anion exchange chromatography containing positively charged diethylaminoethyl (DEAE)-sepharose, facilitating binding of negatively charged heparan sulfate chains. The DEAE column employed measured 2.5 cm × 10 cm. A 50 mL bed volume was equilibrated with a buffer containing: Tris/HCl (0.05 M, pH 8.6), urea (2 M), NaCl (0.25 M), EDTA (2.5 mM), benzamidine (0.5 mM), phenylmethylsulfonylfluoride (PMSF) (0.5 mM) and sodium azide (0.02%, *w/v*). Four volumes of equilibration buffer were passed through the column before the concentrated media was loaded. Concentrated media was passed over the equilibrated column at 4 °C 10 times until all 500 mL of media were loaded. The column then was washed with 4 volumes of equilibration buffer at a flow rate of approximately 1 mL/min, collecting 1.5–2.0 mL fractions. Fractions were followed using absorbance readings at 280 nm. Bound molecules then were eluted using a high salt buffer (same as equilibration buffer but with 0.75 M NaCl). Eluants then were separated by size-exclusion chromatography using a column containing Sepharose CL-4B. Fractions were analyzed by dot blot immunoassay using anti-PLN (A7L6) Ab. Pooled fractions containing PLN were dialyzed in ddH_2_O and centrifuged in a speed vac until the desired concentration was obtained. Enriched PLN solution was separated using HiPrep Heparin FF affinity column to bind heparan sulfate chains and subsequently obtaining purified PLN protein. Enriched fractions were pooled, concentrated, and dialyzed into a final PBS working solution. Purity was assessed by silver stain and samples were aliquoted and stored at −80 °C until use. 

#### 2.4.2. Perlecan Domains I, III and IV

PLN domain (Dm) I, Dm IV-1, Dm IV-2, and Dm IV-3, were produced as described previously [32,35,36]. Briefly, transfected human embryonic kidney, HEK293A cells expressing recombinant proteins were expanded and cultured in hyperflasks (Corning, NY, USA) in Dulbecco’s Modified Eagle Medium (DMEM) (Corning, NY, USA) with FBS (2%, *v/v*), penicillin/ streptomycin (1%, *v/v*), and either puromycin (400 ng/mL) (ThermoFisher Scientific, MA, USA) for Dm IV-1 and IV-2, or Geneticin (G418) (1 mg/mL) (Teknova, CA, USA) for Dm I and IV-3. Full media exchanges were conducted every three days. Inhibitors phenylmethylsulfonylfluoride (PSMF, 0.5 mM) and benzamidine (0.5 mM), and sodium azide (0.02%, *w/v*) were added to conditioned media and filtered through a Polyethersulfone (PES) filter flask (0.2 µm, Corning, NY, USA). Conditioned media was concentrated using a 10,000 Dalton cut-off filter then passed through a nickel-nitrilotriacetic acid (Ni-NTA) agarose resin (ThermoFisher Scientific, MA, USA). The column was first equilibrated with imidazole (10 mM) in PBS, conditioned media was applied, then the column was washed with imidazole (20 mM) in PBS, then eluted with imidazole (300 mM) in PBS. Eluted fractions were pooled, and buffer exchanged with Millipore Amicon Ultra-15 Centrifugal Filter Units (Millipore Sigma, Burlington, MA, USA). PLN Dm-III is composed of three cysteine-free, laminin-like globular domains with alternating laminin epithelial growth factor -like cysteine-rich regions [37]. We designed two Dm-III plasmids using the SnapGene cloning tool (Dotmatics, MA, USA), the first encoding the cysteine free, globular region of PLN Dm III-2 (laminin IV-A2) and the second Dm III-2 (IV-A2) followed by a cysteine-rich laminin EGF-like region (Dm III-2 + cysteine). Each contained an elongation factor-1α promoter and BM40 (from basement membrane protein 40) signal sequence for enhanced secretion, as well as a C-terminal FLAG tag and 6x His-tag for purification (VectorBuilder, Chicago, IL, USA). Plasmids were transfected into HEK293A cells using Lipofectamine 2000 (Life Technologies, Carlsbad, CA, USA). Transfected cells were grown from single-cell clones and selected with G418 (2 mg/mL). Dm III-2 and Dm III-2 + cys production was confirmed via Western blot using 6x His-tag Ab (Invitrogen, Carlsbad, CA, USA). Positive clones were expanded, purified, and sequenced for verification. Conditioned media from hyperflasks was collected and concentrated in bulk using the Sartorius Vivaflow Cross-flow System (Sartorius, Bohemia, NY, USA) with Vivaflow 200 10,000 molecular-weight cutoff (MWCO) PES filters (Sartorius, NY, USA). Dm III-2 and Dm III-2 + cys were purified using Ni-NTA resin as described for Dm IV recombinant proteins with one additional wash of 500 mM NaCl after conditioned medium flow through and before the imidazole (20 mM) wash. As before, the purified protein was buffer exchanged and stored at −80 °C.

#### 2.4.3. Perlecan Domain V

PLN Dm V was produced and purified as previously described [38]. Human Dm V was cloned into the pSecTag2A vector (Invitrogen, Carlsbad, CA, USA) using the primers 5′ Dm V Arthrobacter sp. (AscI) pSecTag2A: 5′-AGGGCGCGCCATCAAGATCACCTTCCGGC-3′; 3′ Dm V Xanthomonas holcicola (XhoI) pSEcTag2A: 5′-AGCTCGAGCCGAGGGGCAGGGGCGTGTGTTG-3′. To confirm that Dm V effects were not due to any single clone-specific irregularities, Dm V was also cloned into the vector pCepPu (provided by Maurizio Mongiat, Center for Cancer Research, Aviano, Italy) using the following primers: Neisseria mucosa heidelbergensis (NheI) whole Dm V forward 5′-AGGCTAGCGATCAAGATCACCTTCCGGC-3′; XhoI HIS Dm V reverse 5′-AGCTCGAGCATGATGATGATGATGATGCGAGG-3′. The Dm V cDNA was amplified from human umbilical vein endothelial cell (HUVEC) cDNA using a GC-rich PCR system and a ready-to-use solution of PCR-Grade Nucleotides, dNTPack (Roche Applied Science, Penzberg, Germany). Maxi-preps of Dm V DNA were transfected into human 293FT (a fast-growing variant of the 293 cell line, for pSecTag2A vector, ATCC) or 293 EBNA (293 cell line expressing the Epstein–Barr nuclear antigen, for pCepPu vector) cells via Lipofectamine (Invitrogen, Carlsbad, CA, USA). After transfection, the 293 cells were put into a CELLine Adhere 1000 bioreactor (Argos Technologies, Vernon Hills, IL USA) and grown for 7 days in complete medium containing FBS (10%, *v/v*), antibiotic/antimycotic, G418 sulfate (1%, *v/v*), and puromycin (0.05 μg). After 7 days the complete medium was removed. Cells then were washed 5 times with CD293 (chemically defined, protein-free) medium containing L-glutamine (4 mM), penicillin/streptomycin (1%, *v/v*), G418 sulfate (1%, *v/v*), and puromycin (0.05 μg) to remove any serum; and then fresh CD293 medium was added to the cells. Cells then were incubated for 7 days, followed by collection of Dm V-containing conditioned medium, and purification of Dm V via its C-terminal 6X His-tag and Ni-ATA agarose beads (QIAGEN, Hilden, Germany) per manufacturer’s protocol. Eluted fractions containing Dm V were combined and dialyzed against PBS and the purity of the resultant Dm V was confirmed via SDS (sodium dodecylsulfate)-PAGE stained with Brilliant Blue G Colloidal, silver stain (FASTsilver Gel Staining Kit, Calbiochem, San Diego, CA, USA) following the manufacturer’s protocol. 

### 2.5. Localized Surface Plasmon Resonance (LSPR) Experiments

The LSPR-based assay was used to delineate the region of each protein necessary for the structural integrity of the matrix-channel tethering complex (M-CTC), which can sensitively monitor the binding interactions between full-length PLN, recombinant subdomains of PLN, and the α_2_ portion of the α_2_δ_1_ subunit. In brief, noble metal nanoparticles display unique localized SPR properties, which are dependent on the size and shape [39,40,41], and most importantly, the dielectric constant of their surrounding environment [42,43]. Utilizing the latter dependency, solid-state, LSPR-based sensors have been developed employing simple optical spectroscopy to detect biological constituents by monitoring the LSPR changes (∆*λ*) induced by their presence [44,45]. Here, we utilized gold triangular nanoprisms (Au TNPs) as plasmonic transducers because they are highly stable, sensitive and receptive to surface modification, and their utility for receptor-analyte binding studies has been previously demonstrated [42]. The surface of Au TNPs was chemically modified for sensing applications and the α_2_ portion of the α_2_δ_1_ subunit attached to the nanoprisms by amide coupling. α_2-_functionalized Au TNPs were then incubated with PLN domains/subdomains, and LSPR wavelengths were measured before and after incubation using UV-visible spectroscopy. The plot of LSPR peak wavelength shift (Δ*λ*) versus PLN concentration in linear scale was used to determine binding affinity (*K_D_*). A schematic representation of LSPR based experiments is summarized in Appendix A and the details of each step are described below.

#### 2.5.1. Synthesis of Gold Triangular Nanoprisms (Au TNPs)

Au TNPs were chemically synthesized according to published procedures [46,47,48]. Briefly, 10.4 mg (0.05 mM) of chloro(triethyphosphine)gold(I) [Et_3_Pau(I)Cl] were dissolved in N_2_ purged acetonitrile (20 mL) and stirred at RT for 5–10 min. Then, 19 µL (0.273 mM) of triethanolamine (TEA) was added to the solution and heated. Upon solution temperature reaching 38 °C, 300 µL of polymethyl-hydrosiloxane (PMHS) was added, and the reaction slowly stirred. During the reaction, the solution color changed gradually from colorless to dark navy-blue, indicating the formation of Au TNPs. Once a dark navy-blue color was achieved, the LSPR dipole peak position (λ_LSPR_) was monitored through UV-visible spectroscopy until the solution was λ_LSPR_ = ~800 nm, indicating the formation of ~42 nm edge length Au TNPs (Appendix A). The Au TNP solution was centrifuged at 7000 rpm for 10 s, transferred to 3-mercaptopropyltrimethoxysilane (MPTMS)-functionalized glass coverslips (Appendix A) and incubated for 1 h. TNP bound coverslips were rinsed with acetonitrile, dried with N_2_ gas, and stored under N_2_ at 4 °C. Au TNP-bound coverslips were used within 3 days of the attachment. 

#### 2.5.2. α2-Functionalized Au TNPs

Au TNP-bound glass coverslips underwent a tape-cleaning procedure to remove non-prismatic structures. Briefly, 3M adhesive tape was placed onto the Au TNP-bound glass coverslip, pressed gently with the thumb, and then the tape was removed at a 90° angle. Cleaned Au TNP-bound coverslips were cut into 6.25 mm × 25 mm pieces using a diamond cutter to produce the sensors (Appendix A). Each sensor was incubated in 6.0 mL of a 1.0 mM:1.0 µM ratio of 11-mercaptoundecanoic acid (MUDA): 1-nonanethiol (NT) solution overnight (Appendix A). The following day, the sensors were rinsed with ethanol to remove loosely bound thiols. This thiol treatment created a self-assembled monolayer (SAM) onto Au TNP surface. Next, SAM-modified Au TNPS were incubated in an EDC/NHS (0.2 M, 1-Ethyl-3-(3-dimethylaminopropyl) carbodiimide/N-hydroxysuccimide) solution for 2 h to activate the acid group of MUDA, rinsed with ethanol and PBS, and incubated overnight in PBS (pH 7.2) containing the α_2_ portion of α_2_δ_1_ (10 ng/mL) (Appendix A). 

#### 2.5.3. Protein Binding Curves and Spectroscopy Characterization

To determine the dissociation constant (*K_D_*) values for interactions between α_2_ and PLN, each α_2_-functionalized sensor was rinsed with PBS and incubated in a solution containing different concentrations (1 × 10^−16^ to 1 × 10^−8^ M) of full-length PLN (digested with heparanase and chondroitinase, or undigested) or each of PLN domains/subdomains Dm I, III-2 (cys free), III-2 (cys), IV-1, -2 and -3 or V (Appendix A). Negative controls were performed by incubating the α_2_-functionalized nanoprisms with PBS devoid of PLN protein. Before and after each incubation step, an extinction spectrum of the sensor was collected through UV-visible spectroscopy, and the shift in the LSPR dipole peak position (Δ*λ*_LSPR_) was obtained (Appendix A). At the end of the experiments, the sensors were removed for data collection. Once we established the regions of PLN that mediate binding within the M-CTC, assays were repeated with the addition of GBP (see drug binding experiments). For UV-vis extinction spectra, λ*_L_*_SPR_ was determined through curve fitting using OriginLab software v9.8 (Northampton, MA, USA). The Δ*λ*_LSPR_ was calculated by taking the difference between the λ_LSPR_ before and after each fabrication step. Values for Δ*λ_LSPR_* in α_2_-sensors (nm) were reported as Means ± standard deviation (SD) of six individual measurements at each concentration of PLN domain/subdomain tested. Absorption and extinction spectra were collected utilizing a Varian Cary 50 Scan UV-visible spectrometer (Varian, Palo Alto, CA, USA) in the range of 300–1100 nm, using 1 cm quartz cuvettes. All spectra were collected in ethanol or PBS (pH 7.2) to keep the bulk refractive index constant. The “background” was a coverslip immersed in ethanol/PBS. The reference (blank) was a sensor incubated in ethanol/PBS (no analyte present). Scanning electron microscopy (SEM) images of Au TNPs were characterized using a JEOL 7800F SEM (JEOL Ltd., Tokyo, Japan).

#### 2.5.4. Data Processing

The statistics software GraphPad Prism version 9.3.1(471) (La Jolla, CA, USA) was used to develop protein binding curves by non-linear least squares fit, plotting Δ*λ*_LSPR_ in α_2_-sensors versus PLN concentration in mol/L (M) (Appendix A). The Hill-Langmuir equation (reported as “specific binding with Hill Slope” within the “receptor binding-saturation binding” models in GraphPad Prism) was used to determine the *K_D_* values between α_2_ and PLN domains/subdomains (Appendix A). Best-fit values for Hill coefficient (h) to measure the degree of cooperativity between α_2_ and PLN, and maximum specific binding (Bmax) for each PLN domain/subdomain were also calculated. R-squared values to assess goodness-of-fit and 95% confidence bands were computed for all protein binding curves. It is worth noting that the Au TNPs used in our LSPR assays have an extremely large sensing range. Therefore, to reach a maximum Δ*λ*_LSPR_, an analyte concentration close to few hundred millimolar would be required. As it is extremely challenging to obtain such large concentrations of purified ligands (PLN domains/subdomains), we were unable to directly determine the highest achievable Δ*λ*_LSPR_. However, we indirectly obtained this value by plotting Δ*λ*_LSPR_ vs. the ligand concentrations tested in linear scale and extrapolating the Δ*λ*_LSPR_ to very high concentrations of ligand. Thus, in our experiments the maximum achievable Δ*λ*_LSPR_ for a given concentration of PLN domain/subdomain (Bmax) was utilized to determine the *K_D_*. 

### 2.6. Drug Binding Experiments

LSPR-based experiments were also used to determine the interactions among α_2,_ PLN, and GBP. Three different approaches were performed. First, to determine if GBP was capable of displacing PLN from α_2_ following binding of PLN to α_2_, the α_2_-functionalized sensors (see Section 2.5.2) were incubated overnight with a 6 mL solution containing full-length PLN (10 nM) or PLN Dm III-2 (100 nM), followed by overnight incubation with GBP (6 mL, 0.33 mg/mL). Second, to determine if PLN could displace GBP from α_2_, the α_2_-functionalized sensors were incubated overnight with GBP (0.33 mg/mL), followed by overnight incubation with full-length PLN (10 nM) or PLN Dm III-2 (100 nM). Lastly, to determine if PLN or GBP had greater affinity for α_2_ when provided equal opportunity to bind, the α_2_-functionalized sensors were incubated overnight in a 6 mL mixture of full-length PLN (10 nM) or PLN Dm III-2 (100 nM), and GBP (0.33 mg/mL). GBP was tested at different concentrations ranging from 0.1–1.1 mg/mL. A concentration of 0.33 mg/mL was selected as this was the lowest GBP concentration needed to fully cover the surface of the α_2_-functionalized sensors at which the Δ*λ*_LSPR_ plateaued. Before and after each incubation step, the sensors were thoroughly rinsed with PBS and the end of the experiments, sensors were removed for data collection and processing as described above.

### 2.7. Docking Models

In silico protein–protein, functional interactions and 3D docking models between the vWFA domain of α_2_δ_1_ and domain III-2 of PLN were simulated with the free web HDOCK [49,50] server (http://hdock.phys.hust.edu.cn/ accessed on 9 February 2022). To develop high confidence homology models of protein structures, multiple sequence alignment was conducted using Clustal Omega (1.2.4) [51] (https://www.ebi.ac.uk/Tools/msa/clustalo/ accessed on 16 October 2021). For PLN, the sequences of the three Laminin-IV A subdomains in PLN Dm III [P98160 residues 538–730 (Dm III-1); 941–1125 (Dm III-2), and 1344–1529 (Dm III-3)] were aligned first. Then, the sequence of PLN Dm III-2 [P98160, residues 941–1125] was selected to be aligned against the sequences of the Laminin IV type A1 (P24043; residues 531–723) and Laminin IV type A2 (P24043; residues 1176–1379) domains of Laminin alpha-2. For the vWFA domain, the sequences of the vWFA domains of human thrombospondin 1 (P07996; residues 316–373), thrombospondin 2 (P35442, residues 318–375) and α_2_δ_1_ (residues 253–430 of CACNA2D1 [P54289]) were used for ClustalO alignment. The amino acid sequences for the vWFA domain of the α_2_ peptide (residues 253–430 of CACNA2D1 [P54289]) and PLN Dm III-2 (residues 941–1125 of HSPG2 [P98160]) were input into the protein fold recognition free web server Phyre2 (Protein Homology/analogY Recognition Engine, v2.0) [52] to obtain structural 3D models using known protein templates. The structures with the higher model confidence (the probability that the match between the input sequence and the template is a true homology) and I.D. value (the percentage identity between the input sequence and the template) were chosen for docking. The protein template information and 3D structures were retrieved from the RCSB protein data bank (https://www.rcsb.org/ accessed on 12 December 2021). For the vWFA domain of α_2_δ_1_, the structure of the vWFA from Catenulispora acidiphila (4FX5) was selected (https://www.rcsb.org/structure/4FX5 accessed on accessed on 15 December 2021). For PLN Dm III-2, the structure of the L4b domain of human Laminin alpha-2 [53] (4YEP) (https://www.rcsb.org/structure/4YEP accessed on 15 December 2021) was used as the best match. In the HDOCK server, PDB files for 4FX5 (vWFA) and 4YEP (PLN Dm III-2) were used to populate the information for receptor and ligand, respectively. The output with the highest docking energy score from the top 10 predictions was selected for visualization.

### 2.8. Animal Experiments and In Vivo Ulnar Loading

Male C57BL/6J mice were purchased from the Jackson Laboratory (JAX, Bar Harbor, Maine) and group-housed (2–4 mice/cage) on Teklad 7099 (TEK)-fresh bedding in ventilated cage systems at the Indiana University School of Medicine animal facilities. Food and water were provided ad libitum and mice were maintained under 12 h light/dark cycles and standard conditions of temperature and humidity. At 16 weeks of age, mice were randomly assigned into 2 groups to receive daily intraperitoneal injections of saline (vehicle, VEH) or gabapentin (GBP, 300 mg/kg BW; 50 mg/mL stock diluted in saline) (Acros Organics AC458020050, ThermoFisher Scientific, MA, USA) for 4 weeks (*n* = 9 mice/treatment). Based on human to mouse dose conversion guides [54], the GBP dose selected for our animal experiments is equivalent to the standard clinical dose in humans of 1800 mg/day (i.e., ~25 mg/kg/d for a 70 kg individual = ~300 mg/kg/d in mice), which is consistent with previous publications in rodents [24,55,56]. Sample size calculations were based on published data to detect histomorphometrically measured changes in bone formation induced by loading of 100 µm^3^/µm^2^/yr, and a true difference between loaded and non-loaded bones as small as 40 µm^3^/µm^2^/yr (α = 0.05 level; power (1–β) = 80%). GBP and VEH treated mice were subjected to axial ulnar compression to induce anabolic skeletal responses as previously described [57]. Briefly, mice were anesthetized under gas isoflurane and the right ulna was loaded using a sinusoidal (haversine) waveform (−2200 µε, 2 Hz, 180 cycles). Mice received one loading bout every other day over a 10-day period, loading order of mice was randomized each time. Left ulnas were used as non-loaded, contralateral controls. To monitor load-induced bone formation, the fluorochromes calcein (10 mg/kg, Sigma-Aldrich, St. Louis, MO, USA) and alizarin (20 mg/kg, Sigma-Aldrich, St. Louis, MO, USA) were injected intraperitoneally one day before the final loading bout and 11 days later, respectively. Mice were euthanized at 20 weeks of age by CO_2_ asphyxiation, followed by cervical dislocation. Ulnas were harvested and processed for dynamic histomorphometry as published [57]. All experiments conducted were approved by the Indiana University Institutional Animal Care and Use Committee. 

### 2.9. Dynamic Histomorphometry

Preparation and histological sectioning of ulnas was conducted by the Histology and Histomorphometry Core within the Indiana Center for Musculoskeletal Health at Indiana University. To detect bone formation changes in double-labeled histological sections, the following parameters were assessed as previously described [57]: periosteal mineralizing surface (MS/BS, %), mineral apposition rate (MAR, μm/day), and bone formation rate (BFR/BS, μm^3^/μm^2^/day). All measurements were collected such that investigators were blinded to treatment. Statistical analyses were conducted using GraphPad Prism). A two-way Analysis of Variance (ANOVA) was conducted to assess the within-subject effect of loading (loaded and control limbs) and between-subject effects of treatment (VEH vs. GBP) as well as interactions between these terms. The Fisher’s Least Significant Difference (LSD) criterion was used for pairwise comparisons between groups. Additionally, the response to mechanical stimuli was calculated for each endpoint as the percent difference between a loaded and non-loaded limb within each animal [%Δ  =  ((loaded limb−control limb)/control limb) × 100%)]. Unpaired Student’s *t*-tests compared the %Δ in the responses to loading between GBP and VEH groups. Results are reported as mean ± standard error of the mean (SEM). Significance level was defined as *p* ≤ 0.05. 

## 3. Results

### 3.1. α2δ1 and Perlecan (PLN) Co-Localize in Murine Osteocyte-like Cells

We conducted double immunostaining to test whether PLN co-localizes with the pore-forming Ca_v_3.2 (α_1H_) VSCC subunit, with wheat germ agglutinin (WGA), and/or α_2_δ_1_ in the murine osteocyte-like cell line, MLO-Y4 [25]. As we previously reported, Ca_v_3.2 (α_1H_) is the primary α_1_ VSCC subunit in osteocytes [28]. In MLO-Y4 cells, Ca_v_3.2 (α_1H_) is expressed within the cell, but also along the cell periphery (Appendix A). As WGA binds N-acetyl glucosamine sugars, which are present on the extracellular α_2_ portion of α_2_δ_1_, we performed double staining with Ca_v_3.2 and WGA-FITC (Appendix A). Areas of overlap (yellow) validated our previous findings that Ca_v_3.2 associates with α_2_δ_1_ in osteocytes (Appendix A). To determine if PLN associates with Ca_v_3.2 (α_1H_) channels, double staining with Ca_v_3.2 (α_1H_) and PLN was performed (Appendix A). Several areas of overlapping signal demonstrated that PLN associates with Ca_v_3.2 (α_1H_) (Appendix A). Consistent with these findings, PLN and WGA staining overlapped in areas along the cell membrane (Figure 2a–c). Immunostaining of MLO-Y4 cells using antibodies specific to α_2_δ_1_ and PLN, demonstrated that both α_2_δ_1_ (Figure 2d) and PLN (Figure 2e) are produced in osteocytic cells. Merged images showed strong overlapping fluorescent signal of these two proteins (Figure 2f, yellow areas). Importantly, the signal was most prominent along osteocytic processes, demonstrating co-localization of PLN and α_2_δ_1_ in the area of greatest mechanosensitivity (Figure 2d–f). 

All cell culture immunostaining assays showed no signal when probed with normal IgG in place of the primary antibodies or when using N, N’, N’’- triacetylchitotriose as a negative control for WGA-FITC staining (Appendix A). Consistent with the immunostaining results, coimmunoprecipitation assays using MLO-Y4 lysates, showed that α_2_δ_1_ and PLN interact forming a complex in vitro (Figure 2g, original blot images in Appendix A). Overall, these data suggest that this matrix (PLN)-channel (α_2_δ_1_) tethering complex is a critical component for mechanosensory responses in osteocytes.

### 3.2. α2δ1 and PLN Bind with High Affinity, Which Is Mediated by PLN Domain III-2

To quantify the molecular interactions between PLN and α_2_δ_1_ we first tested the binding affinity of full-length PLN protein (native form/undigested and enzymatically digested) and the α_2_ portion of α_2_δ_1_, followed by quantifying the binding affinity of individual PLN domains/subdomains (Dm I, III-2, IV-1, IV-2, IV-3, and V) with α_2_. Using Localized Surface Plasmon Resonance (LSPR)-based experiments, we obtained the dissociation constant (the constant describing the drug/receptor interactions at equilibrium) between α_2_-bound sensors and PLN. As dissociation constant (*K_D_*) and affinity are inversely related, the lower the *K_D_* value, the higher the affinity of the molecules tested. UV−vis extinction spectra of Au TNPs after surface modification resulted an LSPR wavelength (λ_LSPR_) = 846.7 nm. Covalent attachment of α_2_ resulted in an λ_LSPR_ = 885.7 nm, (a shift (Δ*λ*_LSPR_) = +39 nm). Addition of PBS to α_2_-functionalized sensors was used as a negative control and resulted in a *λ*_LSPR_ = 886.8 nm, a shift of +1.1 nm, indicating a lack of interaction with α_2_ (Appendix A). The Δ*λ*_LSPR_ obtained after incubation of the α_2_ sensors with each PLN domain/subdomain concentration were used to obtain the *K_D_* values. This information is provided in Appendix A and plotted in Appendix A.

With a *K_D_* of 3.6 × 10^−9^ M, full-length PLN (undigested) bound with high affinity to the α_2_ portion of α_2_δ_1_ (Table 1). Removal of heparan sulfate and chondroitin sulfate groups from PLN (digested) resulted in a *K_D_* of 2.6 × 10^−7^ M. When examining the individual domains/subdomains of PLN, Dm III-2 had the greatest affinity to the α_2_ polypeptide, displaying a *K_D_* of 8.0 × 10^−11^ M. We also tested binding of α_2_ to Dm III-2 containing a cysteine-rich sequence. We found that the *K_D_* value for this domain was 7.7 × 10^−6^, suggesting that binding of PLN to α_2_δ_1_ via Dm III-2 is less likely to be mediated through cysteine rich regions. The *K_D_* values of other PLN domains were, Dm I: 7.7 × 10^−6^, Dm IV-1: 1.4 × 10^−7^, Dm IV-2: 4.3 × 10^−4^, Dm IV-3: 2.8 × 10^−4^, and Dm V: 5.1 × 10^−3^ M. These values each demonstrated moderate to weak binding to α_2_ (Table 1). 

To measure the degree of interaction between α_2_δ_1_ and PLN, we calculated the Hill coefficients (h) from our protein binding curves. This parameter describes ligand binding cooperativity, where the ligand affects the binding of forthcoming molecules to the receptor [58]. In our experiments, we obtained h values ranging from 0.086–0.151 (Appendix A), indicative of negatively cooperative binding. These results suggest that as PLN binds α_2_δ_1_, α_2_δ_1_ affinity to subsequent ligand molecules decreases, and that α_2_δ_1_::PLN binding most likely occurs in a 1:1 ratio. Hill coefficients and maximum specific binding values are provided for the interested reader in Appendix A. 

To evaluate binding of PLN and α_2_δ_1_ in silico, computational 3D protein–protein docking models between the von Willebrand Factor A (vWFA) domain of α_2_δ_1_ (4FX5) and PLN domain III-2 (4YEP) were generated (Figure 3). The quality report for structure accuracy confirmed that the models used for receptor (4FX5, 95.7%) and ligand (4YEP, 100%) had high sequence identities with the input structures, where a sequence ID > 30% is considered reliable. Quality criteria of input protein structures were analyzed by ProQ (protein quality predictor, free web server, v1) [59], a neural network-based method that predicts the quality of a protein model, as measured by LGscore or MaxSub [60]. Suitable scores for these parameters are classified as correct (LGscore > 1.5; MaxSub > 0.1), good (LGscore ≥ 3 to <5; MaxSub ≥ 0.5 to <8) or very good (LGscore ≥ 5; MaxSub ≥ 0.8). Input models for receptor 4FX5 (LGscore = 5.77; MaxSub = 0.428) and ligand 4YEP (LGscore = 5.811; MaxSub = 0.234) were within the appropriate quality ranges for docking modeling. In the HDOCK server, putative binding modes are ranked according to their binding energy scores [49,50]. A more negative score corresponds to strong binding and a less negative, or positive score, corresponds to weak or non-existing binding. The first of the top ten prediction models for 4FX5 and 4YEP scored a docking energy of −272.35 with a confidence of 0.92 (a confidence score > 0.7 = likely to bind; >0.5 and <0.7 = possible to bind; <0.5 = unlikely to bind). Thus, the results from our model indicate strong protein–protein interactions. Cartoon and surface 3D representations of the highest ranked surface binding prediction model are shown in Figure 3.

Analysis of the highest confidence models using HDOCK consistently points to surface exposed hydrophobic regions as the most likely sites of interaction. For instance, in PLN dm III-2, a stretch of amino acids (1019–1024, HSPG2 [P98160]) repetitively appear at the binding interface interacting with hydrophobic regions on the vWFA of α_2_δ_1_. Interestingly, in multiple sequence alignments using the free bioinformatics tool, CLUSTAL Omega, this region is one of the rare regions of the three highly similar laminin IV type A domains that is unique and lacks homology to the other two. This may explain our observations of binding specificity using LSPR.

### 3.3. Gabapentin Interferes with PLN::α_2_δ_1_ Binding

Since PLN Dm III-2 showed the highest affinity for α_2,_ we then used LSPR assays to determine if binding of PLN Dm III-2 or full-length PLN with α_2_δ_1_ is altered by GBP. This was achieved with a series of assays adding either PLN or GBP to the α_2_ peptide bound to the nanoplasmonic sensor. With α_2_ bound to the nanoplasmonic sensor (Δ*λ*_LSPR_ = 39 ± 3.9 nm) (Table 2), full-length PLN was first added, which generated a +14.4 ± 1.5 nm Δ*λ* (900.1 nm), indicative of binding. Subsequent addition of GBP resulted in a −4.1 ± 0.8 nm shift (896 nm), suggesting partial dissociation of PLN from α_2_ in the presence of GBP (Table 2. Exp. 1, Figure 4a). Next, instead of first adding PLN to the nanoprism-bound α_2_ polypeptide, GBP was added resulting in a +5.8 ± 0.6 nm shift (891.5 nm). Binding of GBP, then was followed by addition of full-length PLN which generated a shift of only 0.1 ± 0.1 nm (891.6 nm), indicating an inability of PLN to bind α_2_ in the presence of GBP (Table 2. Exp. 2, Figure 4b). Using a third approach, full-length PLN was pre-incubated with GBP, and this combination then was added to the α_2_-bound nanoprism. Addition of the PLN/GBP mixture resulted in a +2.7 ± 0.6 nm Δ*λ* shift (888.4 nm). As the +2.7 nm shift was similar to the +5.8 nm shift observed with GBP binding α_2_ compared to the +14.4 nm shift found when PLN bound alone, this indicated that in the presence of both GBP and PLN, with equal opportunity to bind, GBP but not PLN bound to the α_2_ polypeptide (Table 2, Exp. 3, Figure 4c). A similar series of experiments were conducted to quantify the interactions between PLN Dm III-2, α_2_, and GBP. Here, α_2_ bound with high affinity to Dm III-2 (Δ*λ*_LSPR_ = + 12.7 ± 1.1 nm), and the addition of GBP interfered with this association (−4.3 ± 0.9 nm shift) (Table 2, Exp. 4,). When GBP was bound to α_2_ prior to addition of PLN Dm III-2 (+5.4 ± 0.7 nm shift), the presence of GBP restricted binding of Dm III-2 to α_2_ (+0.4 ± 0.5 nm shift) (Table 2, Exp. 5), and incubation of α_2_ with a mixture of Dm III-2 and GBP resulted in a shift in the wavelength of +4.9 ± 0.9nm, indicating that GBP bound to α_2_, but not PLN Dm III-2 (Table 2, Exp. 6). Representative LSPR spectra of these experiments are included in Appendix A.

### 3.4. Gabapentin Impairs Bone Mechanosensation In Vivo

To determine the effects of GBP on skeletal mechanosensitivity we examined changes in anabolic bone responses to mechanical loading in mice treated with GBP or saline (vehicle, VEH). At the time of experiment mice in the VEH and GBP groups had body weights of 29.3 ± 0.41 and 29.9 ± 0.43 g, respectively (mean ± SEM). The body weight was not different between groups (*p* = 0.41) and remained stable over the 4 weeks of treatment. Regardless of treatment, mechanical loading induced increases in periosteal mineral apposition rate (MAR) and bone formation rate (BFR/BS) in mice. In VEH treated mice, as expected, dynamic histomorphometry analyses of loaded ulnas revealed a significant increase in periosteal mineralizing surface (MS/BS) (*p* = 0.022) and BFR/BS (*p* = 0.036) while no change was observed in MAR (*p* = 0.302) compared to non-loaded controls (Table 3, Figure 5a,b).

In contrast, GBP treatment resulted in blunted bone mechanosensitivity and impaired bone formation. Mice treated with GBP had no change in MS/BS (*p* = 0.72), MAR (*p* = 0.56) or BFR/BS (*p* = 0.38) following mechanical loading (Table 3, Figure 5a,b). We also evaluated the relative bone changes with loading and found that the adaptive skeletal responses were significantly different between GBP and VEH groups. While VEH mice were able to increase relative MS/BS by 20.1 ± 6.26%, bones from GBP mice change by −0.12 ± 6.97 and these responses were significantly different between the groups (*p* = 0.049). Similarly, in response to loading, animals in the VEH group had an increase in relative BFR/BS of 58.6 ± 17.6% while mice treated with GBP only increase this parameter by 9.29 ± 6.99, and the extent of these responses were significantly different (*p* = 0.047). Relative changes in MAR in response to loading were not different (*p* = 0.35) between the GPB (18.8 ± 5.05%) and VEH (30.37 ± 9.74%) mice (Figure 5c). The final number of animals included in the analysis was *n* = 9 for the VEH-treated mice and *n* = 7 for the GBP treated mice. One animal from the GBP group was euthanized before completion of the experiment, due to a broken ulna during loading, and another mouse from the same group was removed due poor histological quality of control (non-loaded) sections, and thus inability to conduct proper paired comparisons.

## 4. Discussion

Mechanotransduction requires physical coupling of mechanosensory components and the ability of those components to transduce mechanical signals into biochemical responses [61]. Numerous studies have identified molecules that contribute to mechanical signaling within bone such as sclerostin [57,62], connexins [63,64,65], and focal adhesions [30,66,67,68,69]. However, the mechanism by which force is directly transmitted from the bone matrix to the osteocyte cell membrane remains unclear. Likewise, while the presence of transverse TEs in osteocytes has been established [7,10,28,70], the cell membrane molecules to which PLN-containing tethers bind is unknown.

Our hypothesis that PLN binds to the α_2_δ_1_ subunit of VSCCs was formed through several observations. First, various studies showed that VSCCs regulate skeletal mechano-sensitivity [13,71,72]. Second, spatial positioning of α_2_δ_1_ is optimal for interaction with PLN, in that α_2_δ_1_ has a large extracellular region (α_2_) capable of interacting with ligands. And third, the ability of α_2_δ_1_ to regulate gating kinetics of the α_1_ pore of VSCCs [73] made α_2_δ_1_ a strong candidate receptor for PLN binding. Our data showed that PLN matrix tethers bind α_2_δ_1_ with high affinity, connecting the mineralized bone matrix with the osteocyte cell membrane (Figure 6a). 

We previously demonstrated that α_2_δ_1_ modulates mechanically regulated ATP release in osteocytes via its association with Ca_v_3.2 (α_1H_), the predominant α_1_ pore-forming subunit within these cells [27,28]. The extracellular portion (α_2_) of the α_2_δ_1_ subunit is known to be glycosylated with N-acetyl glucosamine sugars. These glycosylation sites are essential for surface expression of α_2_δ_1_ [74] and have high affinity to WGA [74]. In this work we confirmed expression of α_1H_ in MLO-Y4 cells and found that PLN staining independently overlapped with α_1H_ and WGA fluorescent signals at the cell surface of osteocytic cells, suggesting close physical proximity of PLN and the α_1H_ pore and the sugars attached to α_2_δ_1_. In addition, α_2_δ_1_ and PLN co-localize in osteocytic cells along the dendritic processes of osteocytes, the area most sensitive to mechanical force [75]. Furthermore, by quantifying the molecular interactions between the extracellular portion of α_2_δ_1_ and different PLN domains/subdomains, we demonstrated that α_2_δ_1_ and PLN binding is facilitated within the cysteine-free region of PLN Dm III-2, with *K_D_* values in the picomolar range compared to other PLN subdomains, showing *K_D_* values in the nano to millimolar range. As a reference, binding of biotin and avidin is among the strongest non-covalent affinities known [76] with a dissociation constant of about 1.3 × 10^−15^. This aligns with literature reports in which Dm III mediates the binding of other molecules with PLN, including the fibroblast growth factor (FGF)-7 (N-terminal half of Dm III) [77], platelet-derived growth factor (PDGF) (Dm III-2) [78], and FGF18 (Dm III, cysteine-free region) [79]. Further, previous work showed that PLN binds to another matrix molecule called von Willebrand Factor A-domain-Related Protein (WARP) [80]. Notably, the interaction between WARP and PLN is mediated through Dm III-2 of PLN and the von Willebrand Factor A (vWFA) domain of WARP [80]. As the α_2_ portion of the α_2_δ_1_ subunit contains a vWFA domain [81] which enables binding to extracellular matrix molecules [29], these findings provided further reasoning that PLN and α_2_δ_1_ form a functional complex. 

In silico docking models between the vWFA domain of α_2_ and PLN Dm III-2 predicted strong interactions between these molecules. Interestingly, we consistently found a unique amino acid sequence within PLN Dm III-2 interacting with hydrophobic regions of the vWFA domain of α_2_. While not a very large surface area, exposed surface hydrophobic regions frequently are implicated in protein–protein interactions. Although there are limitations in the interpretation of HDOCK results [49,50], the quality results for structure accuracy indicate that the docking predictions obtained are reliable. Our 3D models, combined with the LSPR data, confirm that Dm III-2 is a binding site for α_2_, mediating the interaction of the PLN/α_2_δ_1_ complex. Together, this M-CTC, composed of PLN and α_2_δ_1_, is thus spatially, structurally, and biochemically positioned to activate osteocytes in response to mechanical force (Figure 6a).

Whereas several clinical studies link chronic use of GBP with adverse skeletal side effects, including increased fracture risk [23,82], the molecular mechanisms underlying these effects and whether they occur directly in bone are entirely unknown. We hypothesized that GBP disrupts PLN/α_2_δ_1_ binding, affecting the function of the M-CTC, which may explain the skeletal side effects of this medication. GBP recognizes an Arg-Arg-Arg (RRR) motif within the α_2_ region of the α_2_δ_1_ subunit [21], located upstream and in close proximity to the vWFA domain (Figure 1). Interactions occurring in regions flanking the vWFA can restrict the conformation of the domain (i.e., close, low affinity vs. open, high affinity ligand binding states) [83]. Thus, binding of GBP to the RRR motif may disrupt vWFA-mediated interactions of α_2_δ_1_ with other proteins, such as was demonstrated in a recent study where GBP blocked binding of α_2_δ_1_ and thrombospondins [84]. Here, we demonstrated that GBP interferes with binding of PLN (full-length and Dm III-2) and α_2_δ_1_ in vitro, effectively uncoupling the M-CTC (Figure 6b).

We also showed that acute GBP treatment in mice blunts the anabolic bone responses to mechanical loading. Previous studies have shown that both PLN [10] and α_2_δ_1_ [28] are necessary for mechano-transduction in skeletal cells. Thus, GBP may impair osteocyte mechanosensation by disrupting the function of the PLN::α_2_δ_1_ complex and contribute to the deleterious skeletal effects observed with chronic use of these drugs [22,23,24].

In addition to the changes observed with loading in our dynamic histomorphometry assays, when compared to non-loaded control mice, non-loaded mice treated with GBP had increased MS/BS (Figure 5b). While MAR and BFR/BS were numerically higher in mice treated with GBP, these values were not significantly different. The increased MS/BS suggests that GBP treatment alone has an anabolic effect on bone. However, as other studies have shown that a 12-week course of GBP in rats suppresses bone mass [85] and chronic GBP use increases fracture risk [22] one would expect GBP to impair bone formation at baseline. Our observation of increased MS/BS may be due to inter-individual variability found within inbred mouse lines, or uncoupled bone remodeling occurring in mice treated with GBP. Additionally, as our timeline for this study was shorter (4 weeks), the increased mineralization values observed at this early timepoint may precede later reductions in osteogenic activity that were not captured in the current work. Additional studies to examine the chronic effects of GBP on bone mineralization would be beneficial.

Limitations of this study include the use of only male mice for evaluating the in vivo effects of GBP on bone. Ongoing work is focused on understanding the tissue level impact of GBP in female mice. Furthermore, we did not assess binding of PLN Dm II with α_2_δ_1._ However, as we found that Dm III-2 bound with equivalent affinity to that of full-length PLN, we were confident that the observed binding between the full-size core protein of PLN was mediated through Dm III-2. Additionally, in contrast to that of PLN Dm III and α_2_δ_1_, there are no previous studies that support a potential interaction between PLN Dm II and α_2_δ_1._

Notable strengths of this work included the use of LSPR-based experiments to determine the interactions between PLN and α_2_δ_1_. In this regard, the nanoplasmonic sensors provided reproducible limit of detection at the low zeptomolar range, along with quantitative dissociation constant values (*K_D_*) between biomolecules [42,86] with far greater sensitivity than conventional SPR methods. Additionally, while our interest in the M-CTC lie in osteocyte physiology, it is likely that the function of this complex is conserved across numerous tissues. As such, identification of this novel mechanosensory complex may have a dramatic impact on understanding how other tissues regulate mechanosensation, especially as PLN serves mechanotransduction functions in other cell types [87].

## 5. Conclusions

In summary, this work identified novel interactions between the large heparan sulfate proteoglycan PLN and an extracellular auxiliary subunit of VSCCs. Formation of this complex revealed how the transverse tethers previously identified as force transducers in osteocytes attach to the cell membrane, but also provided a greatly expanded understanding of how VSCCs are capable of being activated by mechanical force. Most importantly, our data demonstrate how GBP may negatively regulate bone remodeling by interfering with osteocyte mechanosensation. Better understanding of the mechanisms by which GBP regulates skeletal mechanotransduction will guide the treatment of patients using these drugs and may lead to the design of precision agents efficacious at their target tissues, but devoid of detrimental skeletal effects.

## Figures and Tables

**Figure 1 biomolecules-12-01857-f001:**
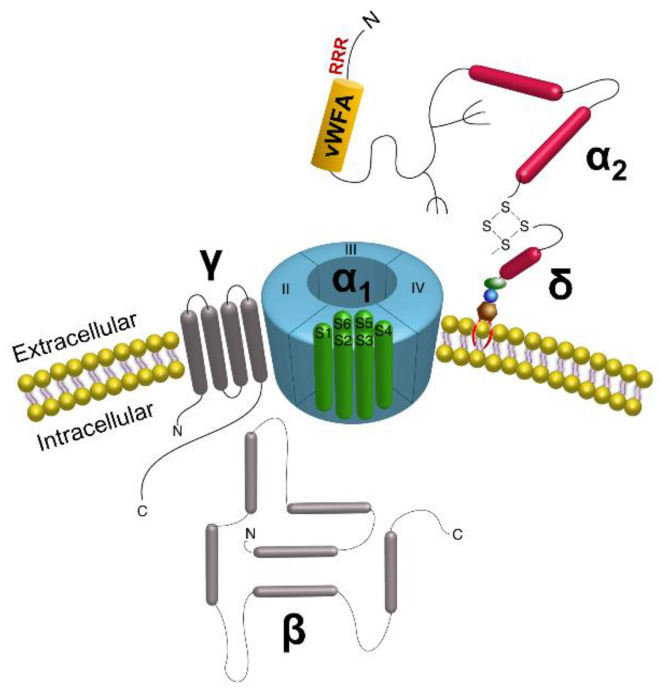
Structure of voltage sensitive calcium channels. The channel complex is composed of the α_1_ pore-forming subunit with auxiliary β, γ, and α_2_δ subunits bound to the pore, positioned to alter gating kinetics of the channel [16,17,18]. The α_2_δ subunit is anchored in the membrane via the δ portion, with the α_2_ region positioned extracellularly. In the extracellular portion (α_2_) of the α_2_δ subunit, the von Willebrand Factor A domain (vWFA) sequence and the Arg-Arg-Arg (RRR) motif for Gabapentin binding are indicated. Reprinted/adapted with permission from Ref. [19], 2022, Springer Nature.

**Figure 2 biomolecules-12-01857-f002:**
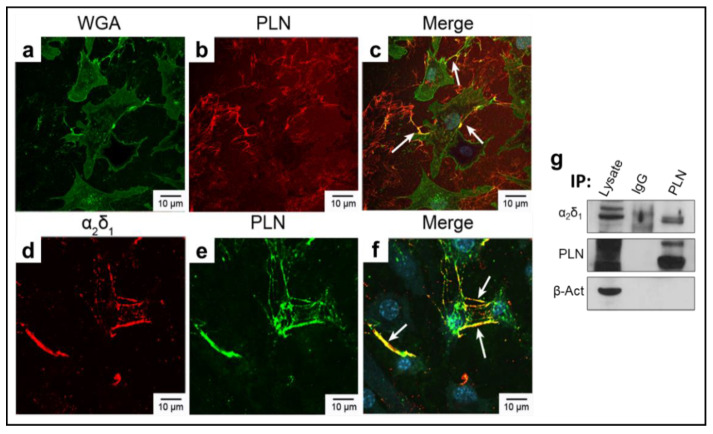
PLN colocalizes with WGA and α_2_δ_1_ in osteocyte-like cells. MLO-Y4 cells stained with (**a**) wheat germ agglutinin (WGA)-FITC (green) and (**b**) perlecan (PLN) (red), (**c**) merge PLN and WGA. On the bottom panels, cells were stained for (**d**) α_2_δ_1_ (red) and (**e**) PLN (green), (**f**) merge PLN and α_2_δ_1_. White arrows indicate overlapping fluorescent signal. (**g**) Co-immunoprecipitation assays from MLO-Y4 lysates show that PLN and α_2_δ_1_ associate. IgG was used as a negative control. Blots were probed for β-actin antibody as a loading control.

**Figure 3 biomolecules-12-01857-f003:**
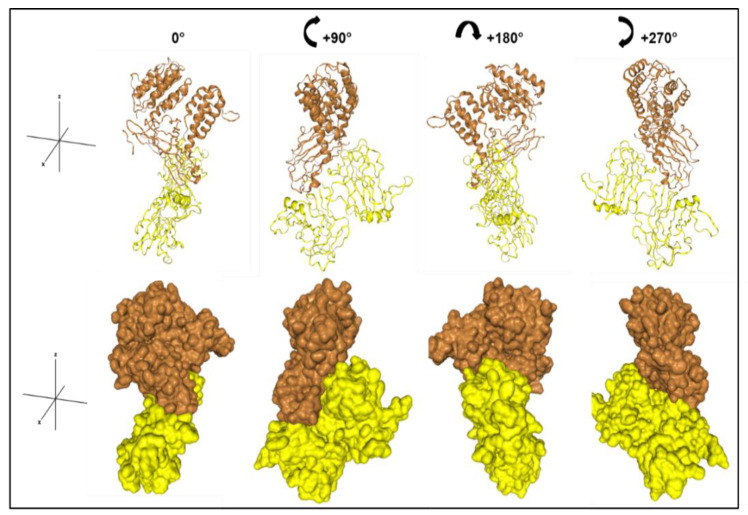
Docking models of vWFA domain of α_2_δ_1_ and PLN Dm III-2. In silico protein–protein functional interactions and 3D docking models between the von Willebrand Factor A (vWFA) domain of α_2_δ_1_ and perlecan (PLN) domain (Dm) III-2 were generated with the free web server HDOCK. 4FX5 (brown) is the vWFA domain and 4YEP (yellow) is the L4b domain of human Laminin α_2_ (PLN Dm III-2). (**Top**), cartoon ribbon-style 3D representations of receptor and ligand. (**Bottom**), surface style representation of the proteins. Each image is rotated 90° clockwise from the previous one.

**Figure 4 biomolecules-12-01857-f004:**
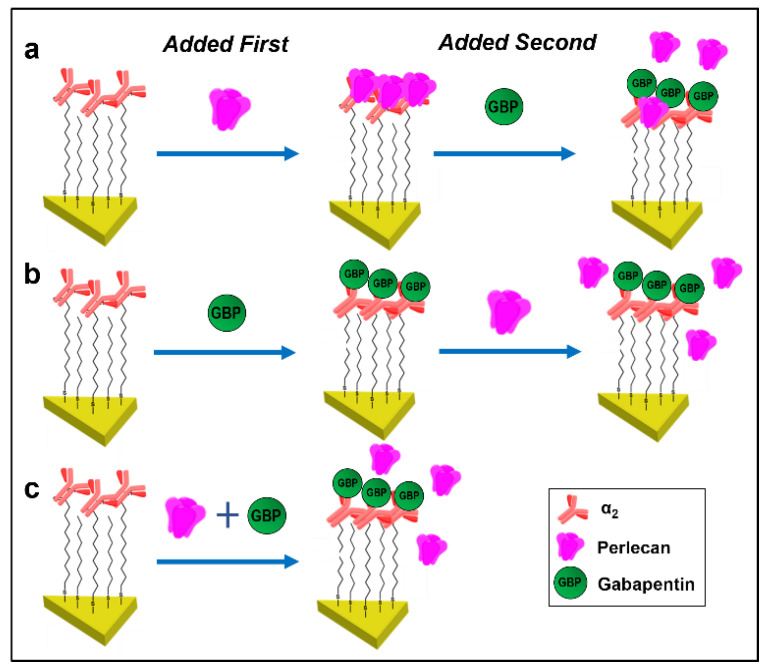
Schematic representation of drug-binding experiments. (**a**) When we first added full-length PLN to α_2_-nanoplasmonic sensors, PLN binds to α_2_. However, further addition of GBP disrupts PLN::α_2_ binding. (**b**) When GBP is added first to α_2_ sensors, further addition of PLN does not disrupt the GBP::α_2_ interaction. (**c**) When addition of PLN and GBP occur simultaneously, α_2_ has greater affinity for GBP.

**Figure 5 biomolecules-12-01857-f005:**
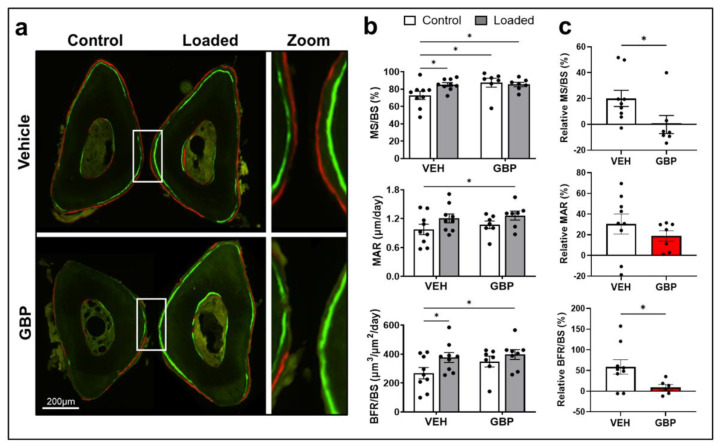
Gabapentin treatment impairs bone mechanosensitivity and load-induced bone formation. Male C57BL/6J mice were injected daily with saline (vehicle, VEH) (n = 9) or Gabapentin (GBP, 300 mg/kg BW) (n = 7) for 4 weeks while undergoing axial ulnar loading. (**a**) Representative images of control (non-loaded) and loaded ulnas from VEH and GBP treated mice. To monitor load-induced bone formation, calcein (green) and alizarin (red) fluorochromes were injected intraperitoneally 8 and 20 days after the first loading day, respectively. Changes in (**b**) mineralizing surface (MS/BS), mineral apposition rate (MAR), and bone formation rate (BFR/BS) in response to mechanical loading were assessed in VEH and GBP treated mice were analyzed by ANOVA and Fisher’s LSD test for pairwise comparisons. (**c**) The percent difference between a loaded and non-loaded limbs within each animal [%Δ  =  ((loaded limb − control limb)/control limb) × 100%)] for MS/BS, MAR and BFR/BS in GBP and VEH groups was calculated analyzed separately by unpaired Student’s *t*-tests. Each dot represents one mouse. Values are shown as Mean ± SEM; (*) *p* ≤ 0.05.

**Figure 6 biomolecules-12-01857-f006:**
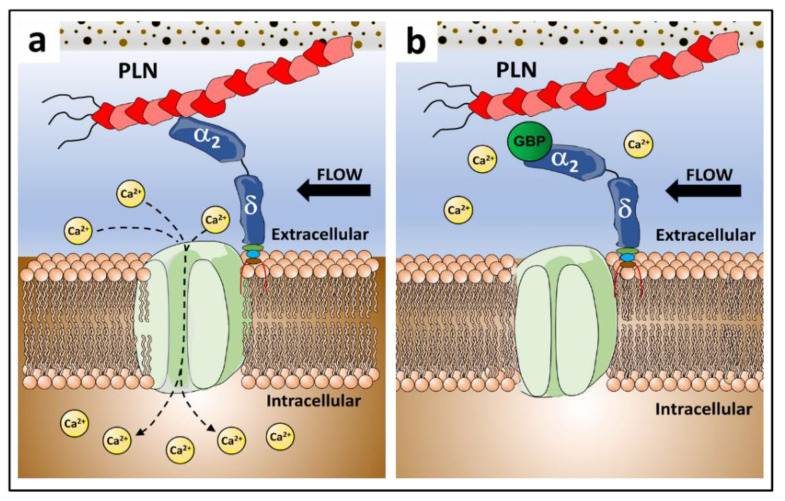
Summary of results. In this work we found that (**a**) the α_2_δ_1_ subunit of voltage sensitive Ca^2+^ channels binds perlecan (PLN) creating a mechanosensory complex that enables connection between the mineralized matrix and the osteocyte cell membrane. (**b**) We also demonstrated that gabapentin (GBP) interferes with binding of PLN and α_2_δ_1_ in vitro. As the PLN/α_2_δ_1_ complex is necessary for mechanotransduction, GBP uncoupling of the complex results in impaired osteocyte mechanosensation in vivo, which may account for the deleterious skeletal effects observed with chronic use of this drug.

**Table 1 biomolecules-12-01857-t001:** Binding affinity experiments between the α_2_ portion of α_2_δ_1_ and perlecan.

Perlecan Domain/Subdomain	*K_D_* (M) ^1^	R-Squared ^2^
Undigested Full Length	3.6 × 10^−9^	0.993
Digested Full Length	2.6 × 10^−7^	0.996
Domain I	7.7 × 10^−6^	0.990
**Domain III-2**	**8.0 × 10** ** ^−11^ **	**0.999**
Domain III-2 (w/cystine)	7.7 × 10^−6^	0.993
Domain IV-I	1.4 × 10^−7^	0.994
Domain IV-2	4.3 × 10^−4^	0.983
Domain IV-3	2.8 × 10^−4^	0.995
Domain V	5.1 × 10^−3^	0.979

^1^ Best-fit values for dissociation constant (*K_D_*). M = Molar concentration. ^2^ R-squared values were used as a measure of Goodness-of-fit.

**Table 2 biomolecules-12-01857-t002:** LSPR-based interactions among α_2_-functionalized sensors, perlecan and gabapentin.

Exp	Sensors	Δ*λ*_LSPR_ (nm)	Added First to Sensors	Δ*λ*_LSPR_ (nm)	Added Second to Sensors	Δ*λ*_LSPR_ (nm)
1	α_2_	+39 ± 3.9	Full length PLN	+14.4 ± 1.5	GBP	−4.1 ± 0.8
2	α_2_	+39 ± 3.9	GBP	+5.8 ± 0.6	Full length PLN	0.1 ± 0.1
3	α_2_	+39 ± 3.9	Full length PLN + GBP	+2.7 ± 0.6	-	-
4	α_2_	+39 ± 3.9	PLN Dm III-2	+12.7 ± 1.1	GBP	−4.3 ± 0.9
5	α_2_	+39 ± 3.9	GBP	+5.4 ± 0.7	PLN Dm III-2	+0.4 ± 0.5
6	α_2_	+39 ± 3.9	PLN Dm III-2 + GBP	+4.9 ± 0.9	-	-

LSPR = Localized surface plasmon resonance, Exp= Experiment, Δ*λ*_LSPR_ = shift in LSPR peak (nm) PLN = Perlecan, GBP = Gabapentin, Dm= Domain. Data is shown as Mean ± Standard deviation.

**Table 3 biomolecules-12-01857-t003:** Ulna changes in response to mechanical loading in vehicle and gabapentin treated mice.

	VEH ^1^ Treated Mice		GBP ^1^ Treated Mice		2-Way ANOVA ^3^		
Bone Parameters ^2^	Non-Loaded Ulna (Ctrl)	Loaded Ulna	Non-Loaded Ulna (Ctrl)	Loaded Ulna	L	T	L×T
MS/BS (%)	72.81 ± 4.7 (9) ^a^	85.23 ± 2.2 (9) ^b^	87.43 ± 5.12 (7) ^b^	85.38 ± 2.51 (7) ^b^	0.192	0.067	0.072
MAR (μm/day)	0.976 ± 0.11 (9) ^a^	1.21 ± 0.09 (9) ^ab^	1.072 ± 0.08 (7) ^ab^	1.26 ± 0.08 (7) ^b^	0.039	0.431	0.431
BFR/BS (μm^3^/μm^2^/day)	268.5 ± 39.13 (9) ^a^	377 ± 34.25 (9) ^b^	348.9 ± 37.24 (7) ^ab^	398.1 ± 34 (7) ^b^	0.041	0.181	0.417

^1^ Mice were injected with saline (VEH, vehicle) or gabapentin (GBP, 300 mg/kg BW) for 4 weeks while undergoing axial ulnar loading. Data are expressed as mean ± SEM (*n*). ^2^ Periosteal mineralized surface/bone surface (MS/BS), mineral apposition rate (MAR), bone formation rate (BFR/BS). ^3^ Two-way ANOVA was conducted to detect loading, drug and interaction main effects; *p*-values for L: Loading T: Treatment and L×T interaction effects. Different letter superscripts denote significant differences among groups, *p* < 0.05, Fisher’s LSD pairwise comparisons.

## Data Availability

All data generated or analyzed during this study are included in this published article and its Appendix A.

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
