# Peer review of "Gabapentin Disrupts Binding of Perlecan to the α2δ1 Voltage Sensitive Calcium Channel Subunit and Impairs Skeletal Mechanosensation"

_biomolecules, 2022, doi:10.3390/biom12121857_

Round 1

Reviewer 1 Report

Perla C Reyes Fernandez et al. have submitted a manuscript entitled “Gabapentin Disrupts Binding of Perlecan to the α2δ1 Voltage 2 Sensitive Calcium Channel Subunit and Impairs Skeletal Mechanosensation” for publication in Biomolecules.

The authors describe a mechananosensitive complex within osteocytes composed of heparan sulfate proteoglycan perlecan (PLN) and the Voltage Calcium channel subunit α2δ1, necessary for bone force transmission and sensitive to the drug Gabapentin.

 The manuscript is concisely written but it can be improved by several points.

 1.       The authors describe that PLN was  isolated and purified from HT-29 human colorectal cancer cells. The functional role in these cell (e.g. versus non transformed cells) might be explained or discussed in more detail.

 2.       Proof of concept: Are there comparable compounds except GBP available as (positive/negative) controls?

 3.       Immunofluorescent stainings: Some antibodies appear to produce a strong background signal (e.g. Cav3.2). Has the specificity tested by e.g. secondary antibodies and/or blocking eperiments?

4.         CoIP: The PLN lane appears to be positive for b-Actin, but in a different height. Please explain.

5.        Impairment of skeletal mechanosensation might be described more carefully, as this relies on a paired comparison of VEH and GBP effects in mice. A more conservative analysis, e.g. multiple comparison including VEH and GBP controls/loaded might be more adequate, here.

 Minor: scale bars are missing in Fig. 5

Author Response

Reviewer #1

General Comments:

Perla C Reyes Fernandez et al. have submitted a manuscript entitled “Gabapentin Disrupts Binding of Perlecan to the α2δ1 Voltage Sensitive Calcium Channel Subunit and Impairs Skeletal Mechanosensation” for publication in Biomolecules. The authors describe a mechanosensitive complex within osteocytes composed of heparan sulfate proteoglycan perlecan (PLN) and the Voltage Calcium channel subunit α2δ1, necessary for bone force transmission and sensitive to the drug Gabapentin. The manuscript is concisely written but it can be improved by several points.

Author Response: We enjoy that the reviewer appreciated the work and its importance in identifying a new mechanosensitive complex in osteocytes. We have made changes that we think improve the presentation in response to reviewer comments.

Specific Comments

Comment 1:

The authors describe that PLN was isolated and purified from HT-29 human colorectal cancer cells. The functional role in these cell (e.g. versus non transformed cells) might be explained or discussed in more detail.

Author Response: We thank the reviewer for the comment. We used the HT-29 human colorectal cancer cell line (previously referred to as WiDr cells) because they secrete large amounts of perlecan essentially as the only large heparan sulfate proteoglycan in conditioned medium (Iozzo, Cell Biol., 99 (1984), pp. 403-417; Fuki et al., J. Biol. Chem., 2000, 18;275(33):25742-50). Our goal here was to isolate full length perlecan to determine binding to α2δ1. Thus, using a cell line that secretes large amounts of PLN was the best strategy to isolate a physiologically relevant form. We used a well-established protocol to isolate PLN, as reported in previous work from our group (Grindel et al, Matrix Biology 2014, 36, 64-76; Tellman et al., IJMS 2021, 22, 3218). We also note that PLN produced from this source in our labs has undergone complete sequencing by mass spec, so we know its exact composition. A summary of the isolation and reason for using these cells is included in the main manuscript.

Comment 2:

Proof of concept: Are there comparable compounds except GBP available as (positive/negative) controls?

Author Response: Excellent question. Gabapentinoids are a class of drug that includes gabapentin (GBP, Neurotin) and pregabalin (PGB, Lyrica). There are other compounds that, like GBP and PGB, are α2δ1 ligands, and were developed as new drugs to treat neuropathic pain including: 1) Mirogabalin Besilate, under the market name Tarlige® (approved only in Taiwan and Korea), 2) Phenibut (PHB), also known as Anvifen, Fenibut, or Noofen (used in Russia, Ukraine, Belarus, and Latvia), 3) PD-02003473, a gabapentin-like analog tested experimentally, and 4) Enacarbil (Horizant). However, these compounds are not yet available/approved in the US (Mirogabalin, Phenibut), have only been tested experimentally (PD-0200347) or are very costly, since therapeutic equivalent is not yet available in the US (Enacarbil). In this study we focused on GBP because it is the most common gabapentinoid, has been on the market the longest, and has been shown in clinical studies to have adverse skeletal outcomes, including increased fracture risk. As GBP and PGB both bind α2δ1, we would anticipate similar outcomes with respect to interfering with PLN binding. However, the purpose of the current work was to first determine if PLN and α2δ1 bind, quantify how tightly they bind, and to determine which region of PLN mediates that binding. We have ongoing work examining the musculoskeletal interactions of different gabapentinoids, which are distinct studies from that presented here.

 Comment 3:

Immunofluorescent stainings: Some antibodies appear to produce a strong background signal (e.g. Cav3.2). Has the specificity tested by e.g. secondary antibodies and/or blocking experiments?

Author Response: This is a great point, and we appreciate the reviewer’s attention to this detail. We have performed several controls to demonstrate the specificity of the antibody staining. First, to block non-specific binding sites, cells were incubated for 1h at RT in a solution blocking buffer containing BSA (3%) and goat serum (10%) (the source species for the secondary antibodies) before incubating with primary Abs. To test non-specific interactions of Cav3.2, α2δ1, and PLN primary antibodies, we conducted control assays 1) using non-immune IgG diluted at concentrations equivalent to primary antibodies and 2) incubation with only secondary antibodies as controls. For WGA-FITC staining, N, N’, N’’- triacetylchitotriose, the sugar to which WGA binds, was used to preabsorb as a negative control. The methods and procedures used for IgGs and N, N’, N’’- triacetylchitotriose controls in place of the primary antibodies are described in the manuscript and showed essentially no immunofluorescent signal (Section 3.1). Additionally, we have expanded on the details provided in the text to explain the proper controls to monitor nonspecific binding in our experiments. The control assays, corresponding to each immunofluorescence image presented in the main manuscript and supplementary materials are now found in supplementary Figures S4 and S5. In addition to the controls conducted for this study, we previously reported similar immunofluorescent assays using the same antibodies (and their controls) as those in this work: Cav3.2  (Shao et al, Developmental Dynamics 2005, 234: 1 p. 54-62); perlecan (Thompson et al, JBMR, 2011. 26:3 pp 618-629); and α2δ1 and Cav3.2 (Thompson et al; JBMR 2011, 26:9 pp 2125-2139) for reference.

 Comment 4:

CoIP: The PLN lane appears to be positive for b-Actin, but in a different height. Please explain.

Author Response: We believe this comment refers to the original uncropped images from the CoIP assays (Figure S4). The reviewer correctly pointed out that in the bottom blot of figure S4, the PLN lane appears to be positive for β-actin, but it is found at a different molecular weight. The 2 bands detected in the β-actin exposed blot for the PLN lane are very likely the light and heavy chains of the IgG isotype of the antibody used for immunoprecipitation (i.e., PLN Ab). The IgG heavy and light fragments appear at ~50-55 kDa and 25-30 kDa, respectively consistent with the molecular weight of the bands observed. The β-actin Ab was used to probe the blot, but because this was a pull down, there is no band showing in the PLN lane at the typical molecular weight of β-actin, indicating that β-actin did not bind PLN. Hence, the secondary Ab had nothing to react against at the expected β-actin molecular weight (42 KDa).

Comment 5:

Impairment of skeletal mechanosensation might be described more carefully, as this relies on a paired comparison of VEH and GBP effects in mice. A more conservative analysis, e.g. multiple comparison including VEH and GBP controls/loaded might be more adequate, here.

Author Response: Our in vivo experiments were designed to detect the dynamic bone responses to mechanical stimuli (i.e., the bone adaption to load variations), and how these adaptive responses are affected by GBP treatment, thus the statistical approach was chosen based on examination of those effects. In that approach we compared the limb that was subjected to loading with the contralateral limb that was not loaded within the same animal. The left and right bones in each group were compared by paired t-test, which is the standard analysis for these types of experiments to detect differences between loaded and control limbs as reported in several publications (Holguin et al, 2016 JBMR 12:2215-226; Sujiyama et al, Bone. 2010 Feb; 46(2): 314–321). However, we agree with the reviewer that a more conservative analysis may be more appropriate to understand the additional effects of GBP in the non-loaded condition. Thus, after careful consideration, we conducted two separate analyses. A two-way Analysis of Variance (ANOVA) was conducted to assess the within-subject effect of loading (loaded and control limbs) and between-subject effects of treatment (VEH vs GBP) as well as interactions between these terms. Additionally, the percent difference between loaded and non-loaded limbs within each animal was calculated [%Δ = ((loaded limb−control limb)/control limb) × 100%)] for bone formation parameters, and the %Δ analyzed separately by unpaired Student’s t-tests. This information and related analyses are now updated in the main manuscript.

Comment 6:

Minor: scale bars are missing in Fig. 5

Author Response: Scale bar has been added to figure 5 as requested by the reviewer.

Reviewer 2 Report

The authors have demonstrated the molecular interactions between α2δ1 and PLN domains. Moreover, authors showed that Gabapentin interfere the binding of PLN to α2δ1 in vitro, and suggested Gabapentin impaired bone mechanosensing in the ulnar loading mouse model.

Major comments:

11) How does the interaction of PLN and α2δ1 regulate cellular activities of osteocytes? Does it produce paracrine factors to regulate osteoblast  or osteoclast activities? 

22) Does α2δ1 express in osteoblast lineage (i.e. progenitors and osteoblasts)? Does α2δ1 regulate those cellular activities? i.e. matrix production, proliferation.

33) In dynamic histomorphometry, GBP treated mice had higher bone formation rate and mineralized surfaces as compared to vehicle treated mice under non-loaded condition. Given that GBP reduced the mechanosensing in the limbs, it would be expected to reduce osteogenic activities under body weight bearing. Please explain the phenotypes.

Author Response

Reviewer #2

General Comments:

The authors have demonstrated the molecular interactions between α2δ1 and PLN domains. Moreover, authors showed that Gabapentin interfere the binding of PLN to α2δ1 in vitro, and suggested Gabapentin impaired bone mechanosensing in the ulnar loading mouse model.

Author Response: We are glad that the reviewer agrees with the major conclusions of our work presented here.

Major comments:

Comment 1:

How does the interaction of PLN and α2δ1 regulate cellular activities of osteocytes? Does it produce paracrine factors to regulate osteoblast or osteoclast activities? 

Author Response: This is a great question, and an area of ongoing investigation in our labs that we hope to answer in detail in future publications. The mechanisms through which the lack of α2δ1 regulates signaling to other bone cell types remains under investigation. We anticipate that the interaction of PLN and α2δ1 alters calcium dependent signaling and downstream actions that occur among the cell types present in bone that regulate bone homeostasis.

Comment 2:

Does α2δ1 express in osteoblast lineage (i.e. progenitors and osteoblasts)? Does α2δ1 regulate those cellular activities? i.e. matrix production, proliferation.

Author Response: We have previously shown [PMID: 16059921] that osteoblasts express both L-type Ca(v)1.2 (alpha(1C)) and T-type Ca(v)3.2 (alpha(1H)) subunits, and expect that they also express the auxiliary subunit α2δ1.  Future work from our group is focused on understanding the mechanisms by which α2δ1 regulates proliferation, differentiation and maintenance of all cells in the osteoblast lineage including osteocytes, and the specific role of this subunit in the documented actions of the channel complex.

Comment 3:

In dynamic histomorphometry, GBP treated mice had higher bone formation rate and mineralized surfaces as compared to vehicle treated mice under non-loaded condition. Given that GBP reduced the mechanosensing in the limbs, it would be expected to reduce osteogenic activities under body weight bearing. Please explain the phenotypes.

Author Response:  

The reviewer is correct in pointing out that the values for GBP treated mice are numerically higher in the non-loaded condition compared to vehicle treated for mineralizing surface and bone formation rate. However, our analysis shows that these values are only significantly higher for mineralizing surface (p=0.013, GBP non-loaded vs VEH non-loaded). This difference could reflect the inter-individual variability found within inbred lines, or uncoupled remodeling occurring in mice treated with GBP (i.e., increase in mineralizing surface accompanied with a greater increase in bone resorption parameters at point in time). Other studies have shown that treatment with GBP (150 mg/kg/day) for 12 weeks lowers bone mass in rats (Kanda et al, 2017, Biol. Pharm. Bull. 40, 1934–1940). As such, we agree that the expected outcome would be lower bone responses at baseline. In this work we did not directly measure bone mass and the time of treatment was shorter (4 weeks). Thus, the higher values in mineralization observed in the GBP group at baseline could precede later reductions of osteogenic activities which were not apparent in these studies due to the length of the intervention.

While these baseline differences between vehicle and GBP treatment are interesting, our primary endpoint here was to determine if GBP altered the anabolic bone response to mechanical loading, where we found that GBP has an inhibitory effect on load-induced bone formation. We were able to clearly capture this phenomenon by comparing the percentage difference between loaded vs non-loaded bones of the same animal [%Δ = ((loaded limb−control limb)/control limb) × 100%)] and showing that the extent of the adaptive responses to mechanical stimuli are clearly diminished in GBP treated mice but not in VEH treated mice. Additional text has been added to the discussion to help the reader better understand the phenotypic observations presented in this work.

Round 2

Reviewer 1 Report

The authors addressed all points accordingly.